# Anti-topological crystal and non-Abelian liquid in twisted semiconductor bilayers

Aidan P. Reddy [1], D. N. Sheng[2], Ahmed Abouelkomsan [1], Emil J. Bergholtz [3] ✉ & Liang Fu [1]

We show that electron crystals compete closely with non-Abelian fractional Chern insulators in the half-filled second moiré band of twisted bilayer $MoTe_2$. Depending on the twist angle and microscopic model, these crystals can have non-zero or zero Chern numbers $C$. The $C = 0$ crystal occurs because contributions to the total Chern number from the full first band (+1) and half-full second band (-1) cancel. This is counterintuitive because the first two non-interacting bands in a given valley have the same Chern number $+1$. For these two reasons, we call this crystal an *anti-topological crystal*. The anti-topological crystal is a novel type of electron crystal that may occur in systems with multiple Chern bands at filling factors $n > 1$.

Moiré superlattices made of twisted semiconductor bilayers host a variety of topological and symmetry-broken electronic phases. In twisted $MoTe_2$ ($t$MoTe$_2$), spatially varying intralayer potential and interlayer tunneling together produce topological minibands[1,2]. At partial filling of the lowest band $n < 1$, Coulomb interactions give rise to Ising ferromagnets[3,4], fractional quantum anomalous Hall (FQAH) states[2,3,5–12], anomalous composite Fermi liquids[9,13,14], and other remarkable phenomena[15–17].

Even more intriguingly, recent theoretical works have proposed a non-Abelian fractional Chern insulator (FCI) at half-filling of the second moiré band in $t$MoTe$_2$[18–22], which corresponds to $n = 3/2$ or $5/2$ depending on the degree of spin polarization. This state arises from a close resemblance between the wavefunctions of the second miniband and those of the first-excited Landau level (1LL), which hosts a non-Abelian state at half-filling[23–26]. However, this 1LL-resemblance seems to depend on microscopic effects such as modifications to the moiré potential landscape at small twist angles[19,21]. Given that the non-Abelian FCI in $t$MoTe$_2$ is sensitive to microscopic conditions, what are its energetic competitors?

It is natural to expect that the proposed non-Abelian FCI competes closely with other phases including electron crystals[27]. For instance, in the 1LL, a slight deviation of the two-body interaction from $1/r$ can replace the non-Abelian state with a composite Fermi liquid or a stripe charge density wave[28]. The competition between topology and crystallization also occurs in the lowest band of $t$MoTe$_2$: while the

FQAH state appears robustly at $n = \frac{2}{3}$, a generalized Wigner crystal with zero Hall conductance is predicted and observed at $n = \frac{1}{3}$[17,11,15].

In this work, we study the phase diagram of $t$MoTe$_2$ at half-filling of the second miniband. We identify commensurate electron crystals with unit cells quadrupled relative to the moiré unit cell. To gain a more complete picture of competing phases, we examine both the adiabatic model[29–31], previously shown to host a non-Abelian FCI phase[18], and the lowest-harmonic continuum model from which the adiabatic model is derived[1]. The first two non-interacting minibands in a fixed valley of both models have the same Chern number +1 throughout the twist angle range we focus on. Yet remarkably, they both host "anti-topological" crystals with vanishing many-body Chern numbers $C = 0$. Further, we find that, in the adiabatic model, two additional crystal phases with $C = 2$ and 1 appear at angles near that at which a $C = \frac{3}{2}$ non-Abelian FCI appears.

We base our conclusions on a combination of band-projected exact diagonalization and all-band Hartree-Fock calculations. The anti-topological crystal shares a quadrupled unit cell in common with the quantum anomalous Hall (QAH) crystal recently proposed in $t$MoTe$_2$[32]. However, the QAH crystal studied in ref. 32 occurs at half-filling of a Chern +1 band and has a total Chern number of +1, whereas the anti-topological crystal occurs at three-halves filling of two Chern +1 bands and has a total Chern number of 0. Our work identifies the anti-topological crystal as a new type of electron crystal distinct from crystals in partially occupied Chern bands studied previously and

---

[1]Department of Physics, Massachusetts Institute of Technology, Cambridge, Massachusetts, USA. [2]Department of Physics and Astronomy, California State University Northridge, Northridge, California, USA. [3]Department of Physics, Stockholm University, AlbaNova University Center, Stockholm, Sweden. ✉ e-mail: emil.bergholtz@fysik.su.se

establishes it as a competitor to the non-Abelian FCI state at half-filling of the second miniband in $t$MoTe$_2$.

Recent experiments report the appearance of a $C = 2$ Chern insulator under a finite magnetic field stemming from $n = 2$, providing evidence that the first and second moiré bands in a given valley have the same Chern number[33]. Meanwhile, an insulating state appears near $n = \frac{3}{2}$ at finite magnetic field. Our theory suggests a possible explanation of the $n = \frac{3}{2}$ insulating state as an anti-topological crystal.

## Results

### Incompressible liquids and crystals in adiabatic model

$t$MoTe$_2$ hosts time-reversal-partner and spin-valley-locked Chern minibands in its $K$ and $K'$ valleys. Here, we begin by studying an approximation to the continuum model[1], the "adiabatic" model[18,29], whose one-body spin/valley projected Hamiltonian takes the form

$$H = \frac{(\boldsymbol{p} + \frac{e}{c}\boldsymbol{A}(\boldsymbol{r}))^2}{2m} + V(\boldsymbol{r}). \quad (1)$$

Here, $\boldsymbol{A}(\boldsymbol{r})$ is an effective vector potential whose corresponding effective magnetic field $\boldsymbol{B}(\boldsymbol{r}) = \nabla \times \boldsymbol{A}(\boldsymbol{r})$ varies with moiré periodicity and averages to one flux quantum per unit cell. The adiabatic model minibands are thus equivalent to Landau levels modulated by a periodic magnetic field and scalar potential that originate from the moiré pattern and enclose one flux quantum per moiré unit cell. Throughout this work, we use the continuum model parameters for $t$MoTe$_2$ reported in ref. 11. We focus on filling of $n = \frac{3}{2}$ of a hole per moiré unit cell and assume full spin (or, equivalently, valley) polarization. An external magnetic field may drive full spin polarization through the Zeeman effect when it does not occur spontaneously. In our exact diagonalization calculations, we use a Coulomb interaction $v(r) = \frac{e^2}{\epsilon r}$ and make a variational approximation by assuming that the lowest spin-↑ miniband is full and constraining the remaining holes to the second spin-↑ miniband. Though inert, the holes in the full lowest band add a Hartree-Fock self-energy $\Sigma(\boldsymbol{k})$ to the second band's one-body dispersion[18].

In Fig. 1a, b, we contrast an example many-body spectrum at $n = \frac{3}{2}$ and an interlayer twist angle of $\theta = 2.6°$ with one at $3°$. The first exhibits a sixfold ground state quasidegeneracy as expected of an FCI state

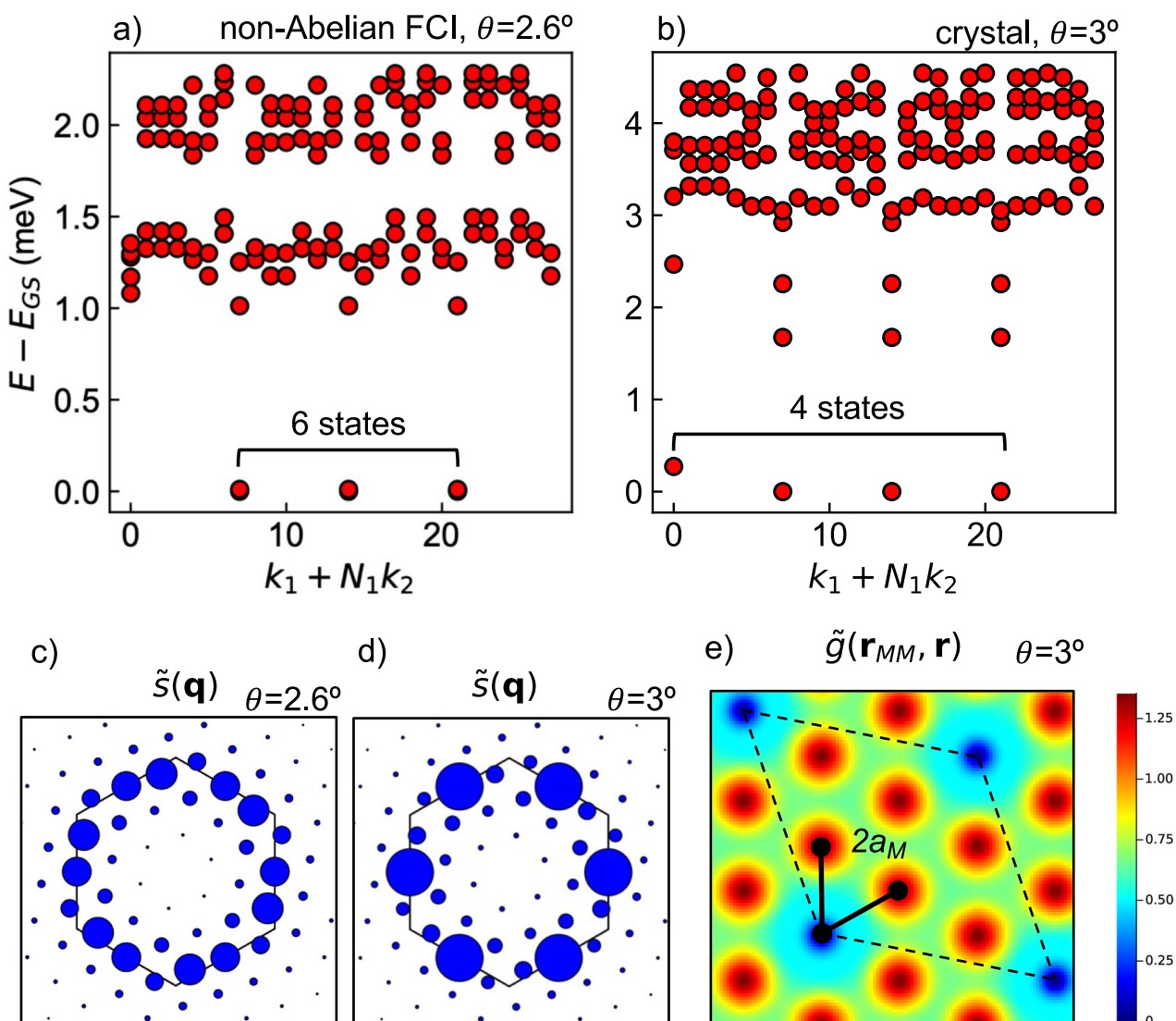

**Fig. 1 | Non-Abelian FCI and electron crystal phases in the adiabatic model.** Representative energy spectra within (**a**) the non-Abelian FCI and (**b**) crystal phases, at $\theta = 2.6°$ and $3°$ respectively. Modified structure factor at (**c**) $\theta = 2.6°$ and (**d**) $\theta = 3°$. The first moiré Brillouin zone is outlined, the radius of the marker is proportional to $\tilde{s}(\boldsymbol{q})$, $\tilde{s}(0) = N$ is omitted, and the maxima of $\tilde{s}(\boldsymbol{q})$ at $2.6°$ and $3°$ are 0.472 and 0.721 respectively. **e** Modified pair correlation function at $\theta = 3°$. A single supercell is outlined. $\epsilon = 5$ and cluster 28 are used. $E_{GS}$ is the ground state energy.

with Pfaffian or anti-Pfaffian topological order and even fermion number[34], while the second exhibits a fourfold ground state quaside-generacy. The center-of-mass crystal momenta of the four ground states at 3° differ by reciprocal lattice vectors of a 2 × 2 enlarged unit cell ($m$ points, or midpoints of the moiré Brillouin zone edges), indicating spontaneous translation symmetry breaking.

To further probe crystallization at $\theta = 3°$, we study density correlation functions. We find that the 1LL-like character of the second miniband's wavefunctions suppresses density fluctuations at $\boldsymbol{q} = m$ and, therefore, masks signatures of crystallization in conventional density correlation functions at finite size. This is a consequence of a zero in the projection of the density operator $e^{i\boldsymbol{q}\cdot\boldsymbol{r}}$ to the 1LL at $|\boldsymbol{q}| \approx |m|$ and is familiar from studies of higher Landau levels[28,35,36]. To expose translation symmetry breaking, we define "modified" correlation functions by replacing the projected density operator of the second band, $\bar{\rho}(\boldsymbol{q}) = \sum_{\boldsymbol{k}} \langle u_{2,\boldsymbol{k}-\boldsymbol{q}} | u_{2,\boldsymbol{k}} \rangle c^{\dagger}_{2,\boldsymbol{k}-\boldsymbol{q}} c_{2,\boldsymbol{k}}$, with $\widetilde{\rho}(\boldsymbol{q}) = \sum_{\boldsymbol{k}} \langle u_{LLL,\boldsymbol{k}-\boldsymbol{q}} || u_{LLL,\boldsymbol{k}} \rangle c^{\dagger}_{2,\boldsymbol{k}-\boldsymbol{q}} c_{2,\boldsymbol{k}}$ where $\langle u_{LLL,\boldsymbol{k}-\boldsymbol{q}} || u_{LLL,\boldsymbol{k}} \rangle = e^{\ell^2(iq\wedge k/2 - q^2/4)}$. Similarly, we define $\widetilde{n}(\boldsymbol{r}) = \sum_{\boldsymbol{k},\boldsymbol{k}'} \psi^{*}_{LLL,\boldsymbol{k}'}(\boldsymbol{r}) \psi_{LLL,\boldsymbol{k}}(\boldsymbol{r}) c^{\dagger}_{2,\boldsymbol{k}'} c_{2,\boldsymbol{k}}$. With these replacements, we define a "modified" projected structure factor

$$\widetilde{s}(\boldsymbol{q}) = \frac{1}{N} \langle \widetilde{\rho}(-\boldsymbol{q}) \widetilde{\rho}(\boldsymbol{q}) \rangle \qquad (2)$$

and modified pair correlation function

$$\widetilde{g}(\boldsymbol{r}', \boldsymbol{r}) = \frac{\langle \widetilde{n}(\boldsymbol{r}) \widetilde{n}(\boldsymbol{r}') - \delta(\boldsymbol{r}, \boldsymbol{r}') \widetilde{n}(\boldsymbol{r}) \rangle}{\langle \widetilde{n}(\boldsymbol{r}') \rangle \langle \widetilde{n}(\boldsymbol{r}) \rangle} \qquad (3)$$

where $N$ is the number of holes in the second miniband and $\langle \rangle$ the expectation value with respect to all degenerate ground states. We fix our gauge such that $\langle \psi_{1LL,\boldsymbol{k}} || \psi_{2,\boldsymbol{k}} \rangle$ is real and positive.

In the putative crystal phase at 3°, $\widetilde{s}(\boldsymbol{q})$ shows incipient Bragg peaks at the $m$ points of the moiré Brillouin zone (Fig. 1a). This is consistent with spontaneous quadrupling of the moiré unit cell. In contrast, in the FCI state at 2. 6°, $\widetilde{s}(\boldsymbol{q})$ is nearly isotropic as shown in Fig. 1c. The modified pair correlation function $\widetilde{g}(\boldsymbol{r}', \boldsymbol{r})$ shown in Fig. 1e also evidences the crystal's 2 × 2 symmetry-breaking pattern. At smaller angles such as $\theta = 2. 3°$, we find behavior in the many-body spectrum and projected structure factor similar to 3° (Supplementary Fig. 1).

Having established a crystal phase, we now study the phase diagram's dependence on interlayer twist. In Fig. 2a, we plot the many-body spectrum as a function of $\theta$. We find three distinct regions of 4-, 6-, and 4-fold ground state quasidegeneracy that correspond to crystal, FCI, and crystal phases respectively. In Fig. 2b, we show $\widetilde{s}(\boldsymbol{q} = m)$ as a function of twist angle. Within the FCI phase ($\theta \sim 2. 6°$), $\widetilde{s}(m)$ is smaller than at other twist angles and $\widetilde{s}(\boldsymbol{q})$ is nearly isotropic (Fig. 1c). Within the crystal phase $\theta \gtrsim 2. 7°$, $\widetilde{s}(m)$ is large and grows with system size, consistent with long-range order in the thermodynamic limit. For $\theta \lesssim 2. 4°$, $\widetilde{s}(m)$ also grows with system size (save for the smallest system studied).

We further characterize the evolution of the quantum ground state by calculating its overlaps with model non-Abelian FCI states. Fig. 2c shows that the overlap between the Coulomb ground state and exact model Pfaffian and anti-Pfaffian states become large in a twist angle window similar to that in which the 6-fold ground state quasidegeneracy appears (Fig. 1a)[18]. The Pfaffian overlap consistently exceeds the anti-Pfaffian overlap, suggesting that the former is more likely to emerge in the thermodynamic limit, in agreement with the conclusions of ref. 18 based on overlaps at $\theta = 2. 5°$. Beyond this twist-angle window, both Pfaffian and anti-Pfaffian ground state overlaps decrease, consistent with the emergence of an electron crystal. We remark that the finite-size clusters

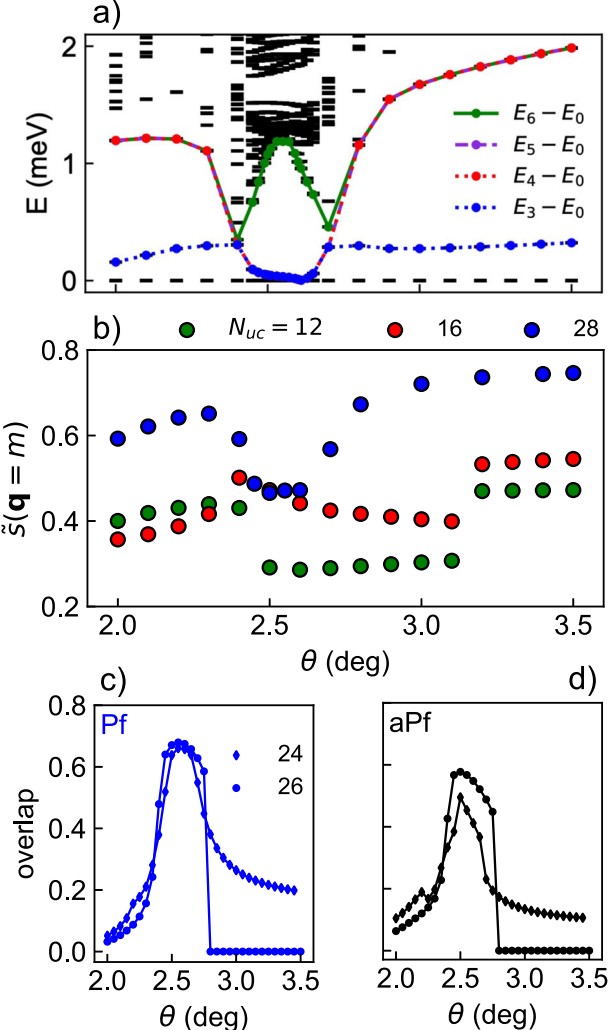

**Fig. 2 | Phase diagram of adiabatic model. a** Low-lying energy spectrum on cluster 28 as a function of twist angle $\theta$. **b** Modified structure factor evaluated at $\boldsymbol{q} = m$ as a function of twist angle $\theta$. Overlap of ground state with the Pfaffian (Pf) (**c**) and anti-Pfaffian (aPf) (**d**) model wavefunctions as a function of $\theta$ on clusters 24 and 26. $\epsilon = 5$ is used.

used may frustrate crystal formation due to incommensurability with a quadrupled unit cell, accounting for the non-zero model state overlap in the putative crystal regions. Our analysis shows that the non-Abelian FCI and crystal phases dominate the window of $\theta$ at hand.

### Crystal in lowest-harmonic continuum model

Having established the broader phase diagram at $n = \frac{3}{2}$ in the adiabatic model, we now turn to the lowest-harmonic continuum model for tMoTe$_2$[1]. We focus on angles $\theta > 1. 9°$ where the first two non-interacting minibands both have $C = +1$ given the parameters of ref. 11. In Fig. 3a, we show the many-body energy spectrum at $n = \frac{3}{2}$ and $\theta = 2. 6°$, where the four lowest energy levels are isolated and nearly degenerate. Their center-of-mass crystal momenta differ by $m$-point wavevectors, consistent with 2 × 2 symmetry breaking. Figure 3b shows that this property holds over a wide range of twist angles 2. 0° ≤ $\theta$ ≤ 2.65°. As in the adiabatic model, we find that signatures of crystallization in density correlation functions are suppressed (Supplementary Fig. 4). When $\theta > 2.65°$, the 4-fold ground state degeneracy

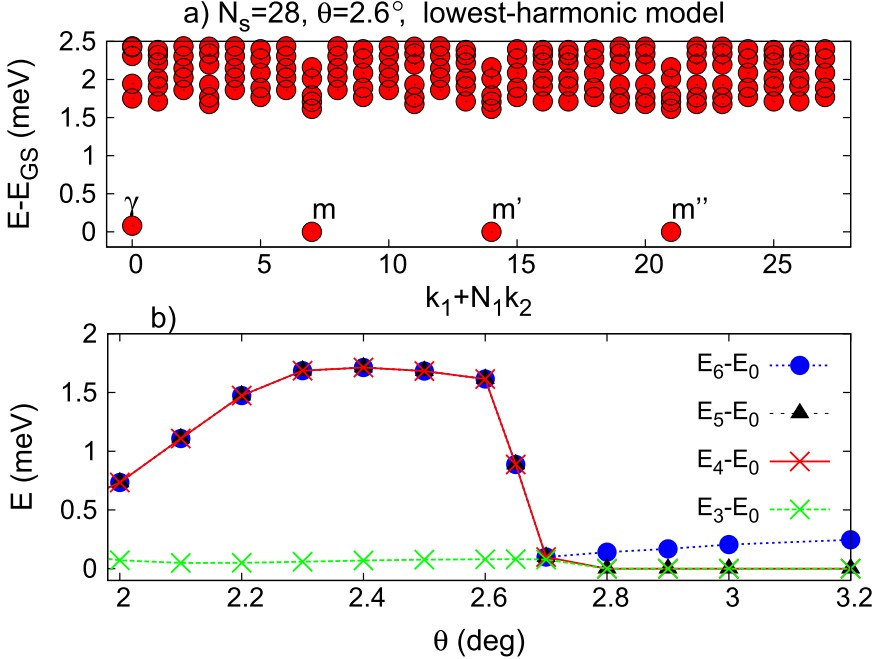

**Fig. 3 | Electron crystal phase from the lowest-harmonic tMoTe$_2$ model. a** Energy spectrum at half-filling of the second moiré miniband for $\theta = 2.6°$. **b** Excitation gaps versus twist angle $\theta$. We use $\epsilon = 5$ and cluster 28. $E_{GS}$ is the ground state energy.

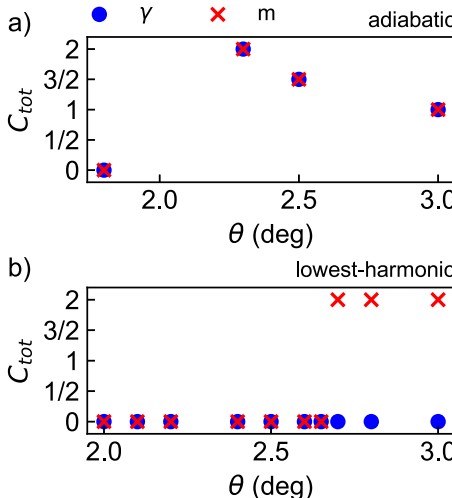

**Fig. 4 | Many-body Chern number.** Total many-body Chern number, $C_{tot} = C_1 + C_2$ (see text) in (**a**) the adiabatic model and (**b**) lowest-harmonic model at several representative twist angles. $n = \frac{3}{2}$ and full spin polarization is assumed. We use $\epsilon = 5$ and cluster 28.

disappears. Instead, we find 6 ground states at low-symmetry momenta, consistent with a Landau Fermi liquid. Unlike in the adiabatic model, we do not find evidence for a non-Abelian FCI state in the lowest-harmonic model.

## Topology and anti-topology of the crystals

We now study the quantized Hall response $\sigma_H = C\frac{e^2}{h}$ of the incompressible electron crystals by calculating their many-body Chern numbers[37,38]. Within our calculation scheme, the total Chern number of the many-body state $C_{tot} = C_1 + C_2$ is the sum of the Chern number $C_1$ of the full first band and the Chern number of the many-body state in the partially occupied second band $C_2$.

First, we address the adiabatic model. As shown in Fig. 4a, we find $C_2 = \frac{1}{2}$ at $\theta = 2.6°$ and therefore $C_{tot} = \frac{3}{2}$ as expected for a Pfaffian fractional Chern insulator. For $1.8° \lessapprox \theta \lessapprox 2°$, we find $C_2 = -1$. However, for $2° \lessapprox \theta \lessapprox 2.4°$ we find $C_2 = +1$ and for $\theta \gtrsim 2.7°$, we find $C_2 = 0$. Our calculations indicate the existence of three distinct crystals with the same $2 \times 2$ enlarged unit cell but different Chern numbers $C_{tot} = 0, 2$, and 1.

We now turn to the crystal in the lowest-harmonic model. As shown in Fig. 4b, we find $C_2 = -1$ for each of the four quasidegenerate ground states in the window $2.0° \leq \theta \leq 2.65°$. The system is, therefore, an anti-topological crystal with $C_{tot} = 0$. In the metallic phase at $\theta > 2.65°$, the finite-size Chern number differs between the lowest-energy states at $\gamma$ and $m$ and are not meaningful in the thermodynamic limit.

To complement our exact diagonalization results, we perform a self-consistent unrestricted Hartree-Fock (HF) study at $n = \frac{3}{2}$ in the lowest-harmonic continuum model of $t$MoTe$_2$. We assume full spin/valley polarization and allow for spontaneous quadrupling of the moiré unit cell. Unlike in our ED study, here we work directly in a plane-wave basis and do not project to any subset of moiré bands. Figure 5a, b show the hole density $n(\mathbf{r})$ and HF band structure of the self-consistent ground state. The hole density has a $2 \times 2$ enlarged unit cell relative to the moiré superlattice. Additionally, the band structure has a gap at the Fermi level as a consequence of spontaneous discrete translation symmetry breaking. $n = \frac{3}{2}$ of a hole per moiré unit cell equates to 6 holes per symmetry-broken unit cell and thus 6 occupied HF bands. The first 4 occupied HF bands originate from the first non-interacting band and thus carry a cumulative Chern number of $+1$. The $5^{th}$ and $6^{th}$ occupied HF bands originate from the second non-interacting band and carry Chern numbers of 0 and $-1$ respectively. Therefore, the self-consistent HF ground state is an anti-topological crystal with a $2 \times 2$ enlarged unit cell and $C_{tot} = 0$, in agreement with our ED study. This demonstrates that the anti-topological crystal is robust against band mixing effects.

Furthermore, the anti-topological crystal persists across a quadratic band inversion that changes the non-interacting Chern number of the partially filled second moiré band from $-1$ to 1 as $\theta$ increases. Fig. 5d shows that the gap $\Delta_{6,7}$ between the highest occupied ($6^{th}$) and

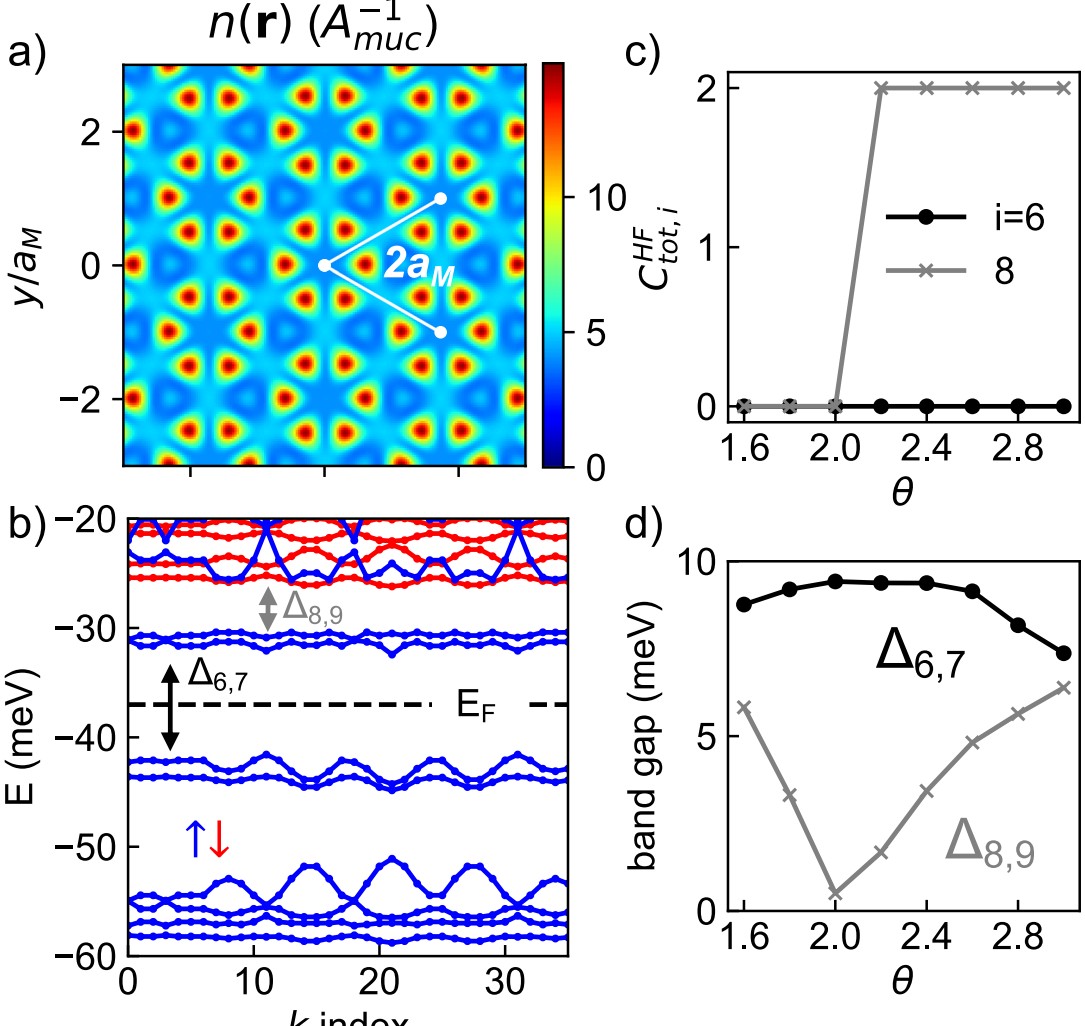

**Fig. 5 | Anti-topological crystal in the Hartree-Fock approximation. a** Hole density $n(r)$ and (**b**) band structure (in the hole picture so the spectrum is bounded from below) of the HF ground state at $n = \frac{3}{2}$ and $\theta = 2.6°$. Blue and red denote the occupied and unoccupied spin/valley respectively. The $k$ index labels crystal momenta in quadrupled unit cell. **c** Cumulative Chern number, $C_{tot,i}^{HF} = \sum_{j=1}^{i} C_j^{HF}$ where $C_j^{HF}$ is the Chern number of the $j^{th}$ symmetry-broken subband, as a function of twist angle. $C_{tot,6}^{HF}$ is the Chern number of the HF ground state at $n = \frac{3}{2}$ and $C_{tot,8}^{HF}$ is the total Chern number of the set of subbands corresponding to the lowest two symmetry-unbroken moiré bands. **d** Minimum direct bandgap between the highest-occupied and lowest-unoccupied subbands $\Delta_{6,7}$, as well as the gap corresponding to that between the second and third symmetry-unbroken moiré bands, $\Delta_{8,9}$. Full spin polarization is assumed. $\epsilon = 20$ and cluster 144 are used.

lowest unoccupied ($7^{th}$) HF bands remains open from $\theta = 2.6°$ to $3°$, while the gap between the eighth and ninth symmetry-broken subbands $\Delta_{8,9}$ closes at $\theta \approx 2.0°$. Moreover, Fig. 5c shows that the cumulative Chern number of the 8 lowest symmetry-broken subbands changes from 0 to 2 at this gap closing. This HF band inversion is related to an inversion between the second and third non-interacting moiré bands at a similar twist angle of $\theta \approx 1.9°$ across which the total Chern number of the lowest two non-interacting bands changes from 0 to 2 (Supplementary Fig. 1). Therefore, the HF ground states on either side of the non-interacting band inversion are both crystals with $C_{tot} = 0$ and are adiabatically connected.

## Discussion

In this work, we have studied novel electron crystals emerging at half-filling of the second miniband in twisted MoTe₂. These crystals appear near in twist angle to a non-Abelian FCI state in the adiabatic model. Although they share a common 2 × 2-enlarged unit cell, they differ in their quantized Hall responses. In the adiabatic model, we find three different crystals with $\sigma_H = 0$, $2\frac{e^2}{h}$, and $\frac{e^2}{h}$. These are all distinct from the

competing non-Abelian FCI with $\sigma_H = \frac{3}{2}\frac{e^2}{h}$. In the lowest-harmonic model, we do not find a non-Abelian FCI state but instead find a crystal with $\sigma_H = 0$. This crystal occurs at half-filling of the second miniband over a range of angles, including angles where the first two non-interacting moiré bands have the same Chern number of $+1$. The contributions to the crystal's total Chern number from its parts in the full first band (+1) and the half-full second band (-1) cancel, leading to $\sigma_H = 0$. Because this crystal exists in two Chern $+1$ bands yet has a Chern number of zero due to canceling contributions, we call it an *anti-topological* crystal.

To gain insight into the origin of the crystal phases, we have studied a toy model for the half-filled second miniband of $t$MoTe₂: the half-filled 1LL in a weak triangular lattice periodic potential. We show with exact diagonalization calculations that the external potential induces a phase transition from a non-Abelian fractional quantum Hall state to a crystal with a 2 × 2 enlarged unit cell (Supplementary Information Section II). When the minima of the potential form a honeycomb lattice, the part of the wavefunction in the 1LL has Chern number $C_2 = 0$ and thus $C_{tot} = 2$. Upon flipping the sign of the potential so that

its minima form a triangular lattice, the part of the ground state in the 1LL is particle-hole conjugated, resulting in $C_2 \to 1 - C_2 = 1$ and $C_{tot} = 1$. Evidently, this model's phase diagram closely resembles that of the adiabatic model in the region 2. $3° \lesssim \theta \lesssim 3°$. Notably, however, it appears not to host an anti-topological crystal. Therefore, while the $C_{tot} = 2$ and 1 crystals in the adiabatic model can be understood as results of adding a weak periodic potential to the 1LL, the anti-topological crystal cannot. This indicates that the anti-topological crystal relies on a stronger deviation from the conventional Landau level limit.

The anti-topological crystal is distinct from electron crystals in partially occupied Chern bands studied previously. For example, the anti-topological crystal differs from reentrant integer quantum Hall states in which a topologically trivial crystal forms on top of an integer quantum Hall state, preserving its quantized Hall conductance. In contrast, the Chern number of the many-body wavefunction of the anti-topological crystal projected to the second miniband is opposite to the band Chern number itself and cancels the contribution from the full first miniband. In this respect, the anti-topological crystal differs from conventional quantum Hall states and previously studied quantum anomalous Hall crystals[32].

Moreover, the anti-topological crystal exists at a half-integer filling factor and has two electrons per symmetry-broken unit cell (in addition to four coming from the full first band). In these ways, it differs from a Wigner crystal near an integer filling factor with one particle per symmetry-broken unit cell. While the anti-topological crystal is topologically trivial, it should exist in proximity to topological states such as a $C = 2$ Chern insulator at $n = 2$. This contrasts a scenario in which the topological character of the underlying moiré bands is nullified by interaction effects at higher filling factors.

## Data availability
The numerical data presented in this study are openly available in the Zenodo repository with the identifier (https://doi.org/10.5281/zenodo.18423460).

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

## Acknowledgements

We thank Xiaodong Xu for insightful discussions. The work at the Massachusetts Institute of Technology was supported by the Air Force Office of Scientific Research (AFOSR) under Award No. FA9550-22-1-0432 and benefited from computing resources provided by the MIT SuperCloud and Lincoln Laboratory Supercomputing Center. A.P.R. was supported in part by grant NSF PHY-2309135 to the Kavli Institute for Theoretical Physics (KITP). D.N.S. was supported by the U.S. Department of Energy, Office of Basic Energy Sciences under Grant No. DE-FG02-06ER46305. A.A. was supported by the Knut and Alice Wallenberg Foundation (KAW 2022.0348). E.J.B. was supported by the Swedish Research Council (2018-00313 and 2024-04567), the Knut and Alice Wallenberg Foundation (2018.0460 and 2023.0256) and the Göran Gustafsson Foundation for Research in Natural Sciences and Medicine. L. F. was partly supported by the Simons Investigator Award from the Simons Foundation.

## Author contributions

A.P.R. and D.N.S. performed exact diagonalization calculations. A.P.R. performed Hartree-Fock calculations. E.J.B. and L.F. supervised the project. A.P.R., D.N.S., A.A., E.J.B., and L.F. contributed to the analysis and writing.

## Funding

## Competing interests

The authors declare no competing interests.
