## [Transparent Peer Review file · Nature Communications]

Anti-topological crystal and non-Abelian liquid in twisted semiconductor bilayers

Corresponding Author: Professor Emil Bergholtz

Version 0:

Reviewer comments:

Reviewer #1

(Remarks to the Author)

The work by Reddy and collaborators reports the discovery of a surprising crystal phase, which the authors term the “anti-topological crystal.” This phase closely competes with the non-Abelian fractional Chern insulator state in the second moiré band of small-angle twisted bilayer MoTe_2 . Remarkably, the anti-topological crystal exhibits a counter-intuitive many-body Chern number that does not align with the naive expectation obtained by summing the band Chern numbers over the filled bands. This observation alone is quite striking and intellectually stimulating. The numerical analysis and models presented by the authors are carried out with care and convincingly demonstrate the existence of distinct parameter regimes. The manuscript is clearly written and easy to read. Even more, the models considered are relevant to current experimental studies of small-angle twisted bilayer MoTe_2 and carry direct implications for understanding its phase diagram. I therefore recommend the manuscript for acceptance, but I would like the authors to address the following points in their revision:

1. Driving mechanism of the anti-crystal phase: What is the underlying mechanism for the emergence of the anti-crystal phase? For instance, why does the anti-crystal with $C_2=0$ (or $C_2=-1$) appear above $\theta > 2.7$ (or $1.8 < \theta < 2$) in the adiabatic model? Are there any significant changes in quantum geometry?
2. Comparison of two non-interacting models: Related to the first point, can the authors comment on the distinct behaviors observed in the two non-interacting models? While both models exhibit anti-crystal phases, the detailed features differ. What is the origin of this difference?
3. Nature of the transition in the adiabatic model: In the adiabatic model, around $\theta \sim 2$, C_2 changes from -1 to $+1$. Could the authors elaborate further on the character of this transition? For example, is it associated with Dirac cones closing and reopening a mass gap?

I recognize that some of the above questions may be difficult to address concretely within numerical calculations such as exact diagonalization (or Hartree-Fock). Nonetheless, providing intuitive (possibly heuristic) explanations would greatly benefit readers and help clarify the underlying physics presented in the manuscript.

Reviewer #2

(Remarks to the Author)

The authors study the interacting twisted MoTe_2 bands at filling $3/2$ and report various many-body phases, including crystallized phases, non-Abelian states, and their competition. They emphasize a new concept introduced in this work—the so-called “anti-topological crystal,” referring to a crystallized phase generated by interactions in two (or more) topological non-interacting bands (particularly of identical Chern numbers). The numerical analysis in this work is adequate and convincing. However, I do not recommend publication in Nature Communications. In my opinion, a more specialized journal such as Communications Physics or Physical Review B would be a better fit.

Few comments below.

(1) Model Justification.

The authors study two models: the adiabatic model (introduced in an earlier work by some of the same authors) and the

lowest harmonic model. However, the motivation for studying these two models is not clearly stated. In the literature (e.g., Nature Communications 15, 4223 and Phys. Rev. Lett. 134, 066601), it is well established that including higher harmonics in the continuum model is crucial for accurately capturing DFT band structures. Why, then, is the lowest harmonic model—or even the further simplified adiabatic model—preferred in this work?

(2) Conceptual advance.

A more crucial criterion stems from the authors' main emphasis—the newly introduced notion of an "anti-topological crystal." In my view, the proposed "anti-topological crystal" does not represent a substantial conceptual advance. The authors have not sufficiently justified the novelty and importance of this so-called "anti-topological crystal phase": Is there a novel mechanism or response? Does it generally compete with non-Abelian states?

Crystalline phases naturally compete with various topologically ordered states in such systems. Moreover, the topology of the crystalline phase is sensitive to the details of the downfolding procedure: downfolding redistributes the Berry curvature and is expected to generate different topological behaviors alongside crystalline ordering. In my assessment, this is an entirely expected outcome—the emergence of trivial crystalline phases from interacting topological Chern bands is neither surprising nor particularly noteworthy.

Reviewer #3

(Remarks to the Author)

In the manuscript titled "Anti-topological crystal and non-Abelian liquid in twisted semiconductor bilayers", Reddy et al. performed numerical study on the correlated electronic states in twisted MoTe₂ at filling 3/2. Twisted MoTe₂ has recently attracted a tremendous amount of research interests because of the discovery of fractional Chern insulators at fractional fillings of the first moire band. Analogy with Landau levels suggests an even more exciting possibility of non-Abelian fractional Chern insulator states at half filling of the second moire band which has been theoretically studied in previous work including one by some of the authors. Competing states at filling 3/2 include crystal states that spontaneously break translation symmetry, and the study of competing crystal states is the major goal of this work. By exact diagonalization of the second moire band based on two different models and Hartree-Fock study based on the continuum model, the authors found a crystal state that has total Chern number zero because the contributions from the first and second moire bands cancel. This is surprising because it occurs even when the first and second single-particle moire bands have the same Chern number, and the authors call this crystal state as an "anti-topological crystal".

The manuscript is well-written and presents some interesting results in an experimentally relevant system. I would suggest the manuscript be published after considering the following comments:

1. The discrepancy between the results from two different but related models should be discussed. Is it because the adiabatic approximation fails? Which results are expected to be more reliable?
2. Aside from presenting numerical results, the underlying physics of the anti-topological crystal should be discussed. Does it form because of some correlations between the two bands or due to some peculiar properties of the second moire band? Can it be generalized to other electron systems with topological bands?
3. There are repetitive statements in the abstract: "This is counterintuitive because the first two non-interacting bands in a given valley have the same Chern number +1" and "Surprisingly, it appears even though the first two non-interacting bands in a given valley have the same Chern number +1".

Reviewer #4

(Remarks to the Author)

Version 1:

Reviewer comments:

Reviewer #1

(Remarks to the Author)

I thank the authors for their time and effort in responding to my questions and comments. Their replies satisfactorily address my concerns and clarify the physical interpretation underlying their numerical results. I also appreciate the additional calculations included in the revised Supplementary Information. I therefore recommend that the manuscript be accepted for publication in Nature Communications.

Reviewer #3

(Remarks to the Author)

The authors have addressed all my comments in the response and revised manuscript. I can recommend the manuscript to

be published in the current form.

Response to referees – NCOMMS-25-46918

Aidan P. Reddy,¹ D. N. Sheng,² Ahmed Abouelkomsan,¹ Emil J. Bergholtz,³ and Liang Fu¹

¹*Department of Physics, Massachusetts Institute of Technology, Cambridge, Massachusetts 02139, USA*

²*Department of Physics and Astronomy, California State University Northridge, Northridge, California 91330, USA*

³*Department of Physics, Stockholm University, AlbaNova University Center, 106 91 Stockholm, Sweden*

(Dated: November 17, 2025)

We are grateful for the positive and constructive feedback. Our responses are highlighted in blue. Changes to the main text are also highlighted in blue.

Summary of changes

- We have added a paragraph to the discussion section describing a toy model for the second miniband in $t\text{MoTe}_2$: the 1LL in a weak periodic potential. We have added a figure to the Supplemental Material to substantiate this discussion. This model exhibits 2×2 crystals with $C_{\text{tot}} = 2$ and 1, but *not* an anti-topological crystal. Therefore, it provides an additional perspective from which to view the phase diagram of the adiabatic model, and demonstrates that the anti-topological crystal relies on significant deviation from the conventional Landau level limit.
- We modified a sentence in the introduction that previously read “We examine both the adiabatic model, previously shown to host a non-Abelian FCI phase, and the lowest-harmonic continuum model from which the adiabatic model is derived.” to “To gain a more complete picture of possible competing phases, we examine both the adiabatic model, previously shown to host a non-Abelian FCI phase, and the lowest-harmonic continuum model from which the adiabatic model is derived.” to clarify our motivation for studying the two models.
- We have removed the sentence “Surprisingly, it appears even though the first two non-interacting bands in a given valley have the same Chern number +1.” from the abstract.
- We added a few words specifying that the non-interacting band inversion between the second and third moiré bands is quadratic and causes the second band’s Chern number to change from -1 to $+1$.

I. REVIEWER #1

The work by Reddy and collaborators reports the discovery of a surprising crystal phase, which the authors term the “anti-topological crystal.” This phase closely competes with the non-Abelian fractional Chern insulator state in the second moiré band of small-angle twisted bilayer MoTe_2 . Remarkably, the anti-topological crystal exhibits a counter-intuitive many-body Chern number that does not align with the naive expectation obtained by summing the band Chern numbers over the filled bands. This observation alone is quite striking and intellectually stimulating. The numerical analysis and models presented by the authors are carried out with care and convincingly demonstrate the existence of distinct parameter regimes. The manuscript is clearly written and easy to read. Even more, the models considered are relevant to current experimental studies of small-angle twisted bilayer MoTe_2 and carry direct implications for understanding its phase diagram. I therefore recommend the manuscript for acceptance, but I would like the authors to address the following points in their revision:

We thank the referee for recognizing that our results are striking, stimulating, and convincingly demonstrated, and that our manuscript is well written. We also thank the referee for recommending the acceptance of our work to Nature Communications.

1. Driving mechanism of the anti-crystal phase: What is the underlying mechanism for the emergence of the anti-crystal phase? For instance, why does the anti-crystal with $C_2=0$ (or $C_2=-1$) appear above $\theta > 2.7$ (or $1.8 < \theta < 2$) in the adiabatic model? Are there any significant changes in quantum geometry?

We thank the referee for this question regarding the mechanism for the emergence of the anti-topological crystal phase. First, we consider the adiabatic model. In this model, the second moiré band maps to the first ($n = 1$) Landau level modulated by a periodic potential and magnetic field with one period per magnetic flux quantum. The angle where the second moiré band is most flat, $\theta \approx 2.5^\circ$, corresponds to a point where these modulations have the weakest effect and effectively vanish. On the two sides of this angle, the modulation changes sign so its potential

FIG. 1. (a) Many-body spectrum of the half-filled 1LL in a weak periodic potential on cluster 28. At small $V_0/\frac{e^2}{\epsilon\ell}$, the system is in a non-Abelian fractional quantum Hall phase with an approximate six-fold ground state degeneracy. As $V_0/\frac{e^2}{\epsilon\ell}$ increases beyond ≈ 0.02 , the system transitions to a crystal phase with a fourfold ground state quasidegeneracy and a 2×2 enlarged unit cell. (b) Many-body spectrum as a function of the total crystal momentum at a representative point in the crystal phase. DUPLICATE OF FIG. 2 OF SUPPLEMENTAL MATERIAL.

minima switch to maxima and vice-versa. A nonzero external potential favors the crystal over the non-Abelian liquid state because the crystal's density modulations allow it to lower its energy by aligning with the external potential. When the sign of the external potential flips, the crystal state gets particle-hole conjugated within the second moiré band, causing C_2 to change by 1, the Chern number of the second moiré band. We have added exact diagonalization calculations for the 1LL in a weak external potential to the supplemental material (reproduced in Fig. ?? in this response), as well as a paragraph to the discussion section of the main text, to illustrate this point. This picture correctly explains the sequence of phases observed in the adiabatic model as θ increases in the vicinity of $\theta \approx 2.3^\circ$: $C = 2$ crystal to non-Abelian FCI to $C = 1$ crystal. Notably, however, the weakly modulated 1LL model appears not to host an anti-topological crystal with $C_{tot} = 0$, indicating that the anti-topological crystal relies on strong deviation from the conventional Landau level limit.

At much lower twist angles $1.8^\circ < \theta < 2^\circ$, the deviation from the ordinary Landau level in the adiabatic model is much stronger. Here, the Berry curvature is less uniform because of proximity to a quadratic band inversion between the second and third minibands. These conditions favor the anti-topological crystal. Since the lowest-harmonic model deviates more strongly from a weakly modulated Landau level across a broader range of twist angles than the adiabatic model, the anti-topological crystal dominates its phase diagram.

2. Comparison of two non-interacting models: Related to the first point, can the authors comment on the distinct behaviors observed in the two non-interacting models? While both models exhibit anti-crystal phases, the detailed features differ. What is the origin of this difference?

We thank the referee for this question. To recap, in the adiabatic model, we observe the sequence of phases: anti-topological crystal $\rightarrow C = 2$ topological crystal \rightarrow non-Abelian FCI with $C = 3/2 \rightarrow C = 1$ topological crystal. In contrast, in the lowest-harmonic model, we observe only the anti-topological crystal phases on the large θ side of the band inversion point.

The origin of this difference is that the second band in the adiabatic model more closely resembles the ordinary $n = 1$ Landau level, which is known to host a non-Abelian FQH state at half filling. In contrast, the lowest-harmonic model deviates strongly from the 1LL across all twist angles. Deviation from the 1LL is required to stabilize the anti-topological crystal phase.

3. Nature of the transition in the adiabatic model: In the adiabatic model, around $\theta = 2^\circ$, C_2 changes from -1 to +1. Could the authors elaborate further on the character of this transition? For example, is it associated with Dirac cones closing and reopening a mass gap?

We thank the referee for this question. We cannot comment on the nature of this phase transition between $C_{tot} = 0, C_2 = -1$ and $C_{tot} = 1, C_2 = 1$ crystals near $\theta = 2^\circ$ from our finite-size calculations. However, we note that while there is no ground state level crossing in the many-body spectrum at zero threaded flux near $\theta = 2^\circ$, there is a ground state level crossing at finite threaded flux that causes C_2 to change (see Fig. 2.)

I recognize that some of the above questions may be difficult to address concretely within numerical calculations such as exact diagonalization (or Hartree-Fock). Nonetheless, providing intuitive (possibly heuristic) explanations would greatly benefit readers and help clarify the underlying physics presented in the manuscript.

FIG. 2. Phase transition from $C_{tot} = 0, C_2 = -1$ crystal to $C_{tot} = 1, C_2 = 1$ crystal at $\theta \approx 2^\circ$. In the momentum sectors hosting the quasidegenerate many-body ground states ($k = 1$ in the top row is the γ point, $k = 8$ in the bottom row is one of the m points), many-body level crossings occur under finite inserted flux near $\theta = 2^\circ$, leading to a change in the many-body Chern number from $C_2 = -1$ to $+1$.

II. REVIEWER #2

The authors study the interacting twisted MoTe_2 bands at filling $3/2$ and report various many-body phases, including crystallized phases, non-Abelian states, and their competition. They emphasize a new concept introduced in this work—the so-called “anti-topological crystal”, referring to a crystallized phase generated by interactions in two (or more) topological non-interacting bands (particularly of identical Chern numbers). The numerical analysis in this work is adequate and convincing. However, I do not recommend publication in Nature Communications. In my opinion, a more specialized journal such as Communications Physics or Physical Review B would be a better fit.

We thank the referee for the feedback on our work. We believe that our work is suitable for publication in Nature Communications. See our detailed responses below.

Few comments below.

(1) Model Justification. The authors study two models: the adiabatic model (introduced in an earlier work by some of the same authors) and the lowest harmonic model. However, the motivation for studying these two models is not clearly stated. In the literature (e.g., Nature Communications 15, 4223 and Phys. Rev. Lett. 134, 066601), it is well established that including higher harmonics in the continuum model is crucial for accurately capturing DFT band structures. Why, then, is the lowest harmonic model—or even the further simplified adiabatic model—preferred in this work?

We thank the referee for this question. In this work, we aim to study trends in the phase diagram across a broad range of twist angles. Including higher harmonics in the moiré potential may enable better fitting to DFT band structures at particular angles, but the exact angle-dependent parameterization of the higher harmonics is unknown. The lowest-harmonic model we use has successfully described much of the experimental phenomenology in twisted MoTe_2 , such as the formation of Abelian FCIs, anomalous composite Fermi liquids, charge density waves, and anomalous Hall Fermi liquids in the first moiré band. We studied the adiabatic model as a point of comparison to the lowest harmonic continuum model because, as shown in our earlier work, it hosts a non-Abelian FCI state. By studying both models, we gain a more complete picture of candidates for competing phases at half-filling of the second miniband.

We have revised a sentence in our introduction to clarify our motivation for studying two models.

(2) Conceptual advance. A more crucial criterion stems from the authors’ main emphasis—the newly introduced notion of an “anti-topological crystal.” In my view, the proposed “anti-topological crystal” does not represent a substantial conceptual advance. The authors have not sufficiently justified the novelty and importance of this so-called “anti-topological crystal phase”: Is there a novel mechanism or response? Does it generally compete with non-Abelian states?

We thank the referee for this feedback. We disagree with the comment on the novelty and importance of the anti-topological crystal. The anti-topological crystal is novel in that it differs from other known electron crystals in Chern bands and Landau level systems. For example, when electron crystals form in Landau level systems, they typically have a many-body Chern number equal to the filling factor rounded to the nearest integer (known as the reentrant integer quantum Hall effect). The anti-topological crystal goes beyond this conventional picture and indicates stronger interplay between potential energy, band topology and interaction. We expect that the anti-topological crystal generally competes with fractional Chern insulator states in systems with multiple Chern bands of the same Chern number, which is highly relevant to experimental studies of multi-Chern-band systems including MoTe_2 .

Crystalline phases naturally compete with various topologically ordered states in such systems. Moreover, the topology of the crystalline phase is sensitive to the details of the downfolding procedure: downfolding redistributes the Berry curvature and is expected to generate different topological behaviors alongside crystalline ordering. In my assessment, this is an entirely expected outcome—the emergence of trivial crystalline phases from interacting topological Chern bands is neither surprising nor particularly noteworthy.

We thank the referee for this feedback. We believe that the anti-topological crystal is distinct from previously studied electron crystals in Chern band systems. We agree with two other referees that our work meets the high standard of the journal and should be published in Nature Communications.

III. REVIEWER #3

In the manuscript titled “Anti-topological crystal and non-Abelian liquid in twisted semiconductor bilayers”, Reddy et al. performed numerical study on the correlated electronic states in twisted MoTe_2 at filling $3/2$. Twisted MoTe_2 has recently attracted a tremendous amount of research interests because of the discovery of fractional Chern insulators at fractional fillings of the first moire band. Analogy with Landau levels suggests an even more exciting possibility of non-Abelian fractional Chern insulator states at half filling of the second moire band which has been theoretically studied in previous work including one by some of the authors. Competing states at filling $3/2$ include crystal states that spontaneously break translation symmetry, and the study of competing crystal states is the major goal of this work. By exact diagonalization of the second moire band based on two different models and Hartree-Fock study based on the continuum model, the authors found a crystal state that has total Chern number zero because the contributions from the first and second moire bands cancel. This is surprising because it occurs even when the first and second single-particle moire bands have the same Chern number, and the authors call this crystal state as an “anti-topological crystal”.

The manuscript is well-written and presents some interesting results in an experimentally relevant system. I would suggest the manuscript be published after considering the following comments:

We thank the referee for recognizing our work as surprising, interesting, experimentally relevant, and well written. We also thank the referee for recommending that our work be published in Nature Communications.

1. The discrepancy between the results from two different but related models should be discussed. Is it because the adiabatic approximation fails? Which results are expected to be more reliable?

We thank the referee for this question. We present the results from both calculations side-by-side to show plausible candidates for ground states at half-filling of the second miniband in twisted MoTe_2 . There is some uncertainty in the best model to use to describe MoTe_2 , so we can not say with certainty which model will describe the system more faithfully.

2. Aside from presenting numerical results, the underlying physics of the anti-topological crystal should be discussed. Does it form because of some correlations between the two bands or due to some peculiar properties of the second moire band? Can it be generalized to other electron systems with topological bands?

We thank the referee for this question. The anti-topological crystal forms due to correlations within the second moiré band. The fact that band mixing is not essential for the anti-topological crystal is demonstrated by the fact that we find it in ED calculations projected to the second band only. We believe that anti-topological crystals will also form in other systems with multiple Chern bands of the same sign Chern number.

We have revised the manuscript to include a discussion of the half-filled 1LL in a weak external potential. This model shows 2×2 crystals with $C_{tot} = 2$ or 0 depending on the sign of the external potential (see Fig. 1, but not an anti-topological crystal. This analysis shows that the anti-topological crystal relies on a significant deviation from the conventional Landau level limit.

3. There are repetitive statements in the abstract: “This is counterintuitive because the first two non-interacting bands in a given valley have the same Chern number $+1$ ” and “Surprisingly, it appears even though the first two non-interacting bands in a given valley have the same Chern number $+1$ ”.

We thank the referee for this helpful comment. We have removed the sentence “Surprisingly, it appears even though the first two non-interacting bands in a given valley have the same Chern number +1.” from the abstract.

IV. REVIEWER #4

We thank the referee for their time and effort in reviewing our work.